# CHIP as a membrane-shuttling proteostasis sensor

Yannick Kopp[1,2], Wei-Han Lang[1,2†], Tobias B Schuster[1,2†], Adrián Martínez-Limón[1,2], Harald F Hofbauer[1,3], Robert Ernst[1,3‡], Giulia Calloni[1,2], R Martin Vabulas[1,2]*

[1]Buchmann Institute for Molecular Life Sciences, Goethe University Frankfurt, Frankfurt am Main, Germany; [2]Institute of Biophysical Chemistry, Goethe University Frankfurt, Frankfurt am Main, Germany; [3]Institute of Biochemistry, Goethe University Frankfurt, Frankfurt am Main, Germany

**Abstract** Cells respond to protein misfolding and aggregation in the cytosol by adjusting gene transcription and a number of post-transcriptional processes. In parallel to functional reactions, cellular structure changes as well; however, the mechanisms underlying the early adaptation of cellular compartments to cytosolic protein misfolding are less clear. Here we show that the mammalian ubiquitin ligase C-terminal Hsp70-interacting protein (CHIP), if freed from chaperones during acute stress, can dock on cellular membranes thus performing a proteostasis sensor function. We reconstituted this process *in vitro* and found that mainly phosphatidic acid and phosphatidylinositol-4-phosphate enhance association of chaperone-free CHIP with liposomes. HSP70 and membranes compete for mutually exclusive binding to the tetratricopeptide repeat domain of CHIP. At new cellular locations, access to compartment-specific substrates would enable CHIP to participate in the reorganization of the respective organelles, as exemplified by the fragmentation of the Golgi apparatus (effector function).

DOI: https://doi.org/10.7554/eLife.29388.001

**\*For correspondence:**
vabulas@em.uni-frankfurt.de

[†]These authors contributed equally to this work

**Present address:** [‡]Institute of Biochemistry, Medical Faculty, University of Saarland, Homburg, Germany

**Competing interests:** The authors declare that no competing interests exist.

## Introduction

Organisms are constantly exposed to environmental variation which, when excessive and requiring adaptation, is called stress. At the cellular level, different stressors can damage proteins affecting their native structure and thus causing loss of function. A prototypic example is the heat shock that leads to protein misfolding, aggregation and cellular toxicity. To keep the cellular proteome in structural and functional balance, called proteostasis, a network of specialized proteins has evolved (*Balch et al., 2008*; *Balchin et al., 2016*). The proteostasis network (PN) is responsible for protein refolding and, in case of the unrepairable damage, degradation. Different classes of molecular chaperones assist during refolding and E3 ubiquitin ligases are involved in degradation function (*Amm et al., 2014*; *Balchin et al., 2016*). Both chaperones and E3 ligases are helped by a number of obligate and optional protein cofactors that increase the specificity, precision and efficiency of the respective reactions.

C-terminal Hsp70-interacting protein (CHIP) is a paradigm example of how protein folding and degradation pathways cooperate during protein quality control (*Arndt et al., 2007*). Initially identified as a cytosolic cochaperone (*Ballinger et al., 1999*), CHIP was shortly afterwards discovered to be a U-box E3 ubiquitin ligase. It was found to engage with HSP70 and HSP90 molecular chaperones which act as adaptors to detect substrate proteins (*Jiang et al., 2001*; *Murata et al., 2001*). CHIP interacts with the chaperones through the N-terminal tetratricopeptide repeat (TPR) domain. The C-terminal tails of chaperones determines the specificity and affinity of this association, which is rather low (*Wang et al., 2011*; *Zhang et al., 2005*). Weak interaction implies dynamic and transient

collaboration between CHIP and the chaperones; however, a detailed atomic and kinetic understanding of the collaboration is missing.

One key question regarding proteostasis is the molecular and functional differences in space and time at different levels of biological organization: organisms, tissues and cells (*Powers et al., 2009*). Considerable progress has been achieved in elucidating signaling and metabolic mechanisms associated with proteostasis changes during organism aging (*Ben-Zvi et al., 2009*; *Taylor and Dillin, 2011*) and differences of PN composition and stress response across tissues (*Guisbert et al., 2013*; *Tebbenkamp and Borchelt, 2010*). At the level of single cells, an obvious strategy to deal with proteostasis load is to accumulate it in distinct subcellular structures and, in the case of duplication, to inherit it asymmetrically in mother and daughter cells (*Escusa-Toret et al., 2013*; *Kaganovich et al., 2008*; *Miller et al., 2015*; *Sontag et al., 2014*). Several cellular locations are used to concentrate damaged proteins, such as the perinuclear endoplasmic reticulum or perivacuolar space (*Kaganovich et al., 2008*). A number of PN components, including molecular chaperones from HSP70 and HSP90 families but also the cochaperones HSP40 and HSP110, were shown to be involved in sorting misfolded proteins to the deposition sites (*Sontag et al., 2014*).

Mechanisms leading to the rearrangement of the cellular architecture during proteostasis stress are less clear. How do organelles change to support the sequestration of a misfolded proteome? Is protein quality control machinery involved in this aspect of the stress response? Here we report our finding that chaperone-binding ubiquitin ligase CHIP has the capacity to sense an acute deficiency of HSP70 in the cytosol and relocalize to membrane-bound organelles enriched in phosphatidic acid and phosphatidylinositol-4-phosphate. We describe the structural elements in CHIP which determine its function as a stress sensor. Finally, with the fragmentation of the Golgi apparatus, we provide an example that the relocalization of PN components is capable of reorganizing the cellular architecture.

## Results

### Chaperone-free CHIP interacts with cellular membranes and phospolipids

Shortly upon exposing EGFP-CHIP-transfected murine embryonic fibroblasts to 43°C, cells with membranous CHIP localization could be identified (*Figure 1a*, *Figure 1—figure supplement 1a*). CHIP association with membranes was lost if the samples had been fixed with paraformaldehyde before microscopy indicating a dynamic nature of the interaction. EGFP localization did not change during heat shock (*Figure 1—figure supplement 1b*). To exclude metabolic effects of heat shock and to implicate temporal deficiency of molecular chaperones as a reason for CHIP relocalization to cellular membranes, we next used inhibitors of CHIP-interacting chaperones. At 37°C, acute inhibition of HSP70 and HSP90 with VER-155008 and 17-AAG, respectively, leads to the appearance of EGFP-CHIP, but not of EGFP, at the plasma membrane (*Figure 1—figure supplement 1c*) - similar to that taking place during heat shock. CHIP relocalization to membranes showed some specificity and was not induced by arsenite treatment (*Figure 1—figure supplement 1d*).

Misfolded and aggregating polypeptides recruit heat shock proteins (HSPs) along with other components of the PN for shielding, refolding or degradation (*Balchin et al., 2016*). Because the HSP-CHIP association is rather weak (*Assimon et al., 2015*), we reason that sequestration of chaperones with aggregating proteins could lead to the appearance of chaperone-free CHIP *in vivo*, at least temporarily until more HSPs accumulate following transcriptional upregulation. Indeed, transiently transfected CHIP partially lost its association with HSC70 during acute heat shock as determined by immunoprecipitation (*Figure 1b*, *Figure 1—figure supplement 1e*). To detect chaperone-free endogenous CHIP in membranes, subcellular fractionation combined with chemical crosslinking was performed. In preliminary experiments with recombinant protein, the conditions to capture CHIP in its naturally dimeric state (*Nikolay et al., 2004*) were established (*Figure 1—figure supplement 1f,g*). CHIP association with molecular chaperones is determined mainly by its N-terminal tetratricopeptide repeat (TPR) domain interacting with last several amino acids of HSP70 or HSP90 (*Wang et al., 2011*; *Zhang et al., 2005*). When added to cellular lysates, the C-terminal HSP70 octapeptide GPTIEEVD dissociated high-molecular-weight species (CHIP-chaperone complexes) leading to the appearance of an HSP-stripped CHIP dimer (*Figure 1c*, lane 8). Presence of a crosslinked

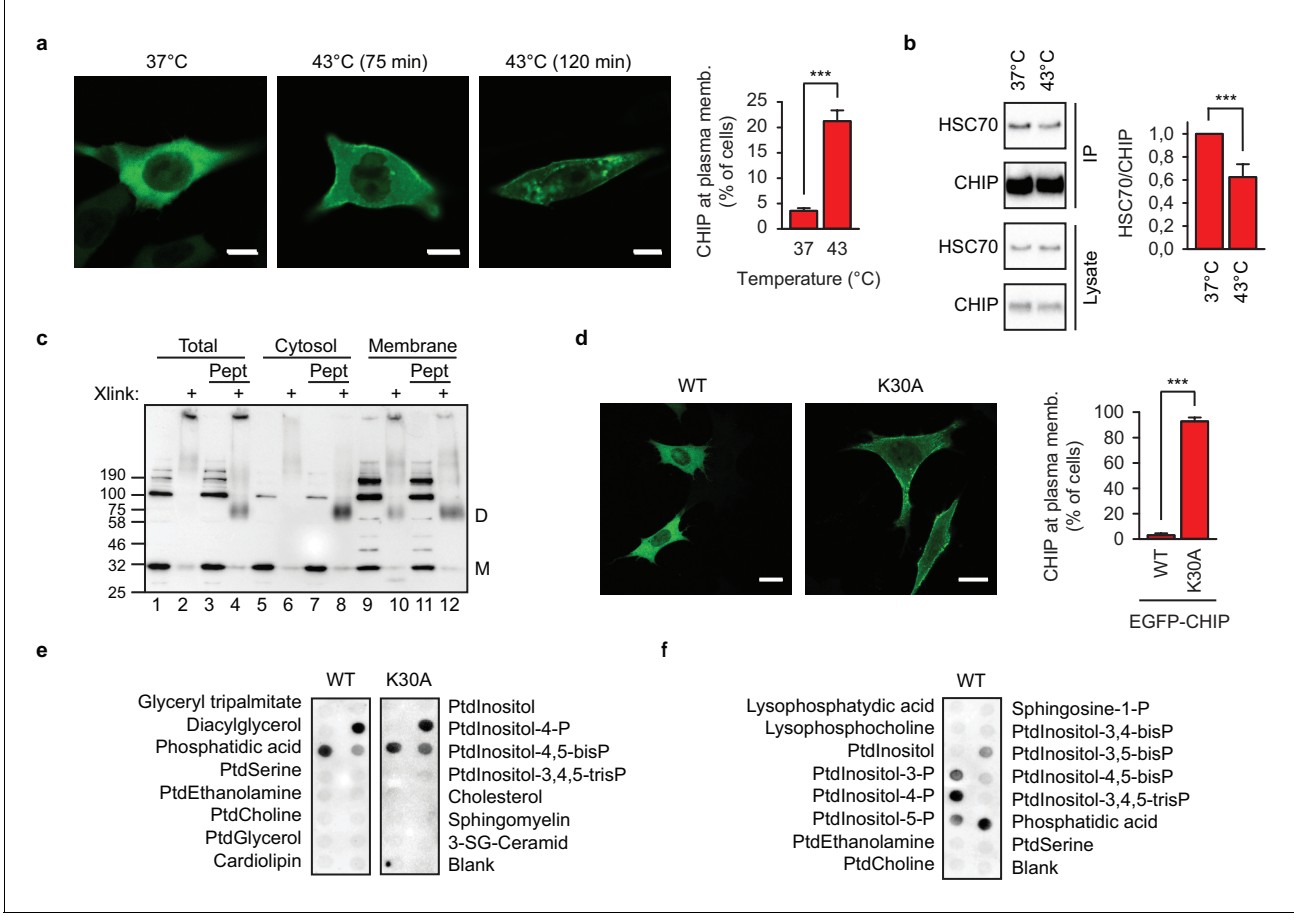

**Figure 1.** Chaperone-free CHIP interacts with cellular membranes in murine embryonic fibroblasts and a distinct set of phospholipids *in vitro*. (**a**) Heat shock (43°C) mobilizes transiently transfected EGFP-CHIP for binding to membranes. Scale bar 10 μm. ***p<0.001, chi-square analysis; N = 3 independent experiments (mean ± SD). (**b**) CHIP loses its association with HSC70 during acute heat shock, 43°C for 60 min, as determined by immunoprecipitation (mean ± SD). ***p<0.001, t-test analysis; N = 4 independent experiments. One representative western blot is shown. (**c**) A fraction of endogenous CHIP in membranes exists in chaperone-free dimeric state as determined by subcellular fractionation and chemical crosslinking. Xlink, cross-linked samples; Pept, C-terminal octapepptide from HSP70; M, monomer; D, dimer. One representative western blot out of three independent experiments is shown. (**d**) K30A mutant CHIP resides at the cellular membranes (mean ± SD). Scale bar 10 μm. One representative out of three independent experiments is shown. ***p<0.001, chi-square analysis; N = 3 independent experiments. (**e**) Chaperone-free wild-type CHIP and K30A variant bind to a set of cellular phospholipids as determined by lipid-binding assay. Ptd, phosphatidyl; P, phosphate. One representative out of three independent experiments is shown. (**f**) Phosphatidylinositol monophosphates are recognized by wild-type chaperone-free CHIP as determined by lipid-binding assay. One representative out of three independent experiments is shown.

DOI: https://doi.org/10.7554/eLife.29388.002

The following figure supplement is available for figure 1:

**Figure supplement 1.** Chaperone-free CHIP association with cellular membranes.

DOI: https://doi.org/10.7554/eLife.29388.003

dimer, even in the absence of the chaperone-competing peptide, in the membrane fraction (*Figure 1c*, lane 10) indicated that endogenous membrane-associated CHIP was at least partially chaperone-free.

To further investigate the possibility that it is the chaperone-free CHIP that interacts with cellular membranes, we used a TPR domain mutant K30A with impaired binding to chaperones. Isothermal titration calorimetry using octapeptide GPTIEEVD confirmed the weak binding to wild-type CHIP ($K_d$ = 5 μM when measured at 37°C) which was further reduced in case of K30A mutant (*Figure 1—figure supplement 1h*). As predicted, a membranous localization of EGFP-CHIP-K30A was observed when compared to its wild-type version (*Figure 1d*). The difference could not be explained by different steady-state levels of the proteins upon transient transfection (*Figure 1—figure supplement 1i*).

To exclude localization differences due to altered lipid binding specificity, we probed the respective recombinant proteins on lipid arrays *in vitro*. Chaperone-free CHIP and the K30A variant showed a similar binding pattern thus excluding accidental formation of a novel determinant of membrane association by the mutation. (*Figure 1e*). Finally, analysis of a broad spectrum of cellular phospholipids revealed CHIP specificity mainly to phosphatidic acid (PA) and phosphatidylinositol monophosphates (PIP), especially PI4P (*Figure 1f*). Binding to phosphatidylinositol-bisphosphates was reduced and there was no binding to phosphatidylinositol-3,4,5-trisphosphate detected, arguing against a trivial electrostatic interaction but pointing instead to high stereospecificity (*Sudhahar et al., 2008*).

In summary, we found that chaperone-free CHIP was able to interact with cellular membranes *in vivo* and a distinct set of phospholipids *in vitro*.

## Membranes and chaperones compete for TPR domain of CHIP *in vitro*

To analyze the details of the interaction, we reconstituted CHIP association with membranes *in vitro* using the recombinantly purified chaperone-free wild-type protein and 1,2-dioleoyl-*sn*-glycero-3-phosphocholine (DOPC) liposomes. A liposome floatation assay revealed cofloatation of CHIP which was strongly enhanced when liposomes were supplemented with 20% PA (*Figure 2a*). CHIP also interacted with liposomes in a liposome pelleting assay, in which pelleting was increased in the presence of PA and also by 1% PI4P, but to a lesser extent (*Figure 2b*). Increase of PI4P to 5% strongly enhanced association of liposomes with CHIP (*Figure 2—figure supplement 1a*). On the other hand, 5% PA was insufficient to strengthen binding of CHIP onto liposomes (*Figure 2—figure supplement 1a*). Notably, CHIP associated with 20% PA- or 5% PI4P-containing liposomes very

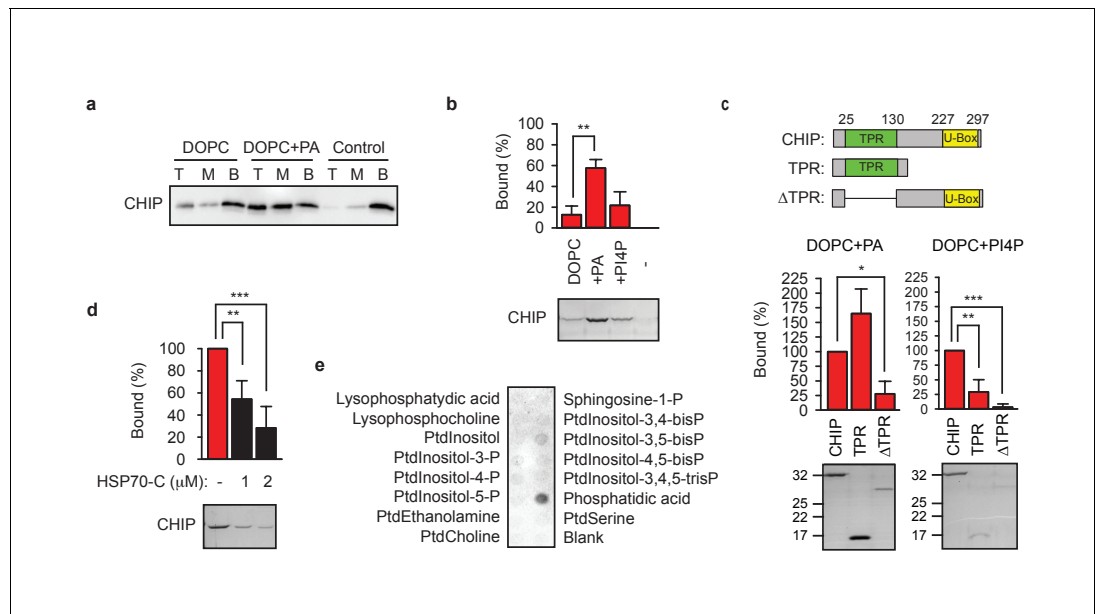

**Figure 2.** TPR domain mediates CHIP association with membranes. (a) Phosphatidic acid (PA) enhances CHIP association with DOPC-liposomes in floatation assays. T, M, B, top, middle, bottom of the gradient, respectively. One representative western blot out of three independent experiments is shown. (b) PA strongly and phosphatidylinositol-4-phosphate (PI4P) weakly enhance CHIP association with DOPC-liposomes in pelleting assay (mean ± SD). **p<0.01, t-test analysis; N = 3 independent experiments. One representative Coomassie Blue-stained gel is shown. (c) TPR domain mediates binding of CHIP to indicated liposomes in pelleting assay (mean ± SD). Schematic depiction of protein variants used to test binding to liposomes indicates N-terminal TPR domain and C-terminal U-Box of CHIP. ***p<0.001, **p<0.01, *p<0.05, t-test analysis; N = 3 independent experiments. Representative Coomassie Blue-stained gels are shown. (d) C-terminal HSP70 (260 amino acid fragment, HSP70-C) affects association of CHIP with liposomes during liposome pelleting (mean ± SD). ***p<0.001, **p<0.01, t-test analysis; N = 4 independent experiments. One representative western blot is shown. (e) HSP70-C in ten-fold molar excess blocks interaction of CHIP with phospholipids as determined by lipid-binding assay. One representative out of three independent experiments is shown.

DOI: https://doi.org/10.7554/eLife.29388.004

The following figure supplement is available for figure 2:

**Figure supplement 1.** TPR domain-lipid interaction *in vitro*.

DOI: https://doi.org/10.7554/eLife.29388.005

efficiently, reaching up to 50–60% of input values, which allowed us to use Coomassie Blue-stained SDS-PAGE gels as readout.

The lipid array analyses in *Figure 1e,f* were performed with the antibody recognizing the C-terminal epitope of human CHIP (251–268 aa). Surprisingly, the antibody recognizing the N-terminal epitope (18–37 aa) failed to detect CHIP bound to lipids (*Figure 2—figure supplement 1b,c*). Given the comparable sensitivities of the C-terminal and N-terminal antibodies when tested with recombinant CHIP (*Figure 2—figure supplement 1c*), the result suggested that the N-terminal part of CHIP, the tetratricopeptide repeat (TPR) domain, must be involved in the interaction with lipids thereby hindering its recognition by the respective antibody. TPR domains are found in numerous proteins where they usually mediate protein-protein interactions (*Zeytuni and Zarivach, 2012*). However, there is at least one example of a TPR domain-lipid interaction as in the case of protein phosphatase 5 binding fatty acids (*Chen and Cohen, 1997*). To test the assumption that the TPR domain from CHIP has dual – protein and lipid - specificity, we cloned and purified the isolated TPR domain and TPR domain-free CHIP (*Figure 2c*, upper panel outlining the deletions). We decided to include the neighboring N- and C-terminal regions while preparing the recombinant TPR domain in order to preserve its functional and structural integrity. According to deuterium exchange measurements, the N-terminal tail is very dynamic extending this flexibility over the first two α-helices in the TPR domain (*Graf et al., 2010*); from the other side, 16 C-terminal α-helical amino acids seem to cap the last tetratricopeptide repeat (*Zhang et al., 2005*). The truncated proteins remained soluble and did not pellet during ultracentrifugation (*Figure 2—figure supplement 1d*). The liposome cosedimentation analyses confirmed that the TPR domain is required for the interaction with membranes containing either PA or PI4P (*Figure 2c*). Notably, the isolated TPR domain bound to the PA-containing liposomes more strongly than the parental full-length protein reaching ca. 160% efficiency of the full-length protein. The relatively high PA concentration (20%) in liposomes seems not to be a reason of this effect, because the TPR domain still surpassed the full-length CHIP in binding capacity when liposomes with lower concentration (5%) of PA were tested (*Figure 2—figure supplement 1e*). The high flexibility and asymmetry of CHIP dimers (*Graf et al., 2010*; *Zhang et al., 2005*) possibly influence the dynamics of membrane interaction (e.g. on- and off-rates) which might differ in case of the isolated TPR domains. The argument of dynamic interaction fits well with our microscopy data where fixation of the samples were found to affect the membrane localization of chaperone-free CHIP.

Having established an *in vitro* model, we sought to test directly the notion that the abundance of molecular chaperones regulates CHIP association with membranes. The 260 amino acid-long C-terminal domain of HSP70 (HSP70-C) was used instead of the full-length protein to exclude N-terminal ATPase effects. Surprisingly, even the substoichiometric amount of 1 µM HSP70-C efficiently prevented CHIP from interacting with liposomes (*Figure 2d*). Preincubation of CHIP with an equimolar amount of HSP70-C blocked the association of CHIP with liposomes to ca. 30% of the control. Likewise, incubation of CHIP with lipid arrays in the presence of HSP70-C resulted in complete loss of binding to phosphatidylinositol monophosphates and only residual interaction with phosphatidic acid (*Figure 2e*).

In summary, the TPR domain was observed to mediate the binding of CHIP to liposomes *in vitro*. Molecular chaperones binding to TPR compete with lipids for this interaction efficiently.

## A positively charged patch is required for CHIP binding to phospholipids

Because TPR domain-less CHIP still showed residual binding to phosphatidic acid-containing liposomes, we assumed additional lipid-interacting determinants. Basic amino acid motifs in polypeptides often play an important role in recognition of negatively charged phospholipids (*Stahelin et al., 2014*). Inspection of the CHIP sequence revealed two patches of positively charged amino acids which we named m1 (143–146 aa: KKKR) and m2 (221–225 aa: KRKKR) (*Figure 3—figure supplement 1a*). Both motifs are outside of the TPR domain and are evolutionary conserved (*Figure 3a*, *Figure 3—figure supplement 1a*). When m1 was mutated to alanines, the specificity of lipid binding and association with cellular membranes was not affected (*Figure 3b,c*, *Figure 3—figure supplement 1b*). However, partial exchange of m2 with alanines impaired CHIP-lipid interaction *in vitro* (*Figure 3b*) and efficiently released chaperone-free EGFP-CHIP-K30A from membranes in living cells (*Figure 3c*). Contrary to wild-type CHIP, m2 mutant was strongly present not only in the cytosol, but in the nucleus as well. The positively charged patches m1 and m2 seem to be very similar and they

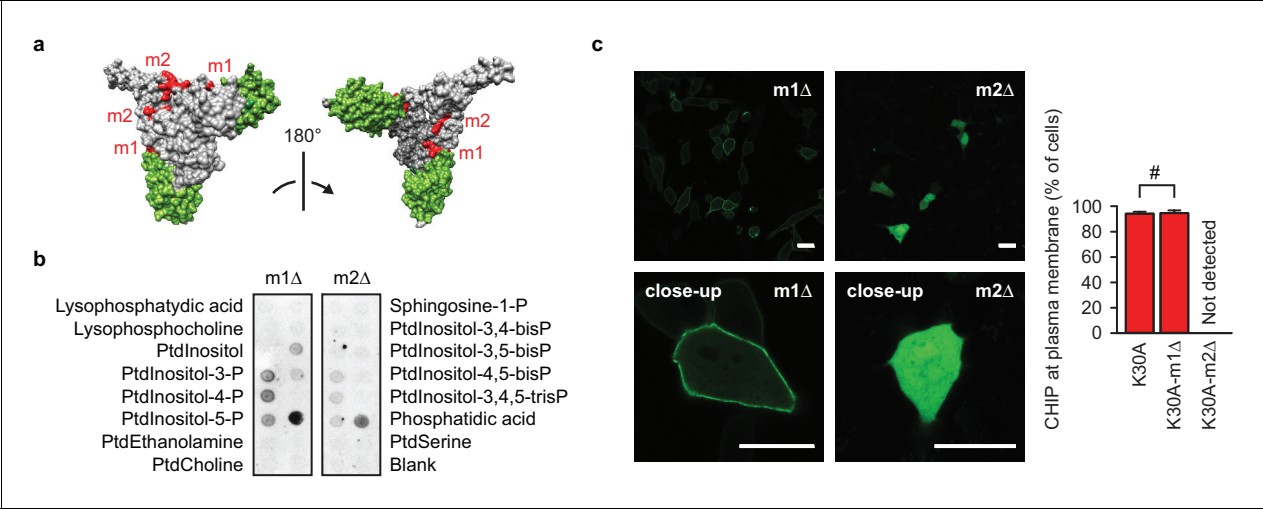

**Figure 3.** A positively charged patch is required for CHIP binding to phospholipids. (a) Surface representation of CHIP dimer (PDB entry 2c2l) with TPR domains colored green. m1, m2, positively charged patches colored red. (b) m2-mutation (m2Δ) affects CHIP interaction with phospholipids *in vitro* as determined by lipid-binding assay. One representative out of three independent experiments is shown. (c) m2Δ CHIP loses association with membranes *in vivo*. Scale bar 20 μm. #, no statistically significant difference according to chi-square analysis; N = 3 independent experiments (mean ± SD).

DOI: https://doi.org/10.7554/eLife.29388.006

The following figure supplement is available for figure 3:

**Figure supplement 1.** Positively charged patches in CHIP are conserved.

DOI: https://doi.org/10.7554/eLife.29388.007

are spatially in close proximity as judged from the crystal structure (*Figure 3a*). Therefore, it is likely that the neighboring amino acids contribute to the striking difference in the subcellular localization when m1 and m2 mutants are compared. An alternative explanation would be the different flexibility of the patches. When compared to m1, m2 is localized in a more dynamic part of the middle domain that fluctuates between unfolded coil and α-helical conformation (*Graf et al., 2010*). At this stage, it is unclear whether coil or α-helical structure contributes to the interaction of CHIP with phospholipids.

A closer look at the asymmetric CHIP dimer (PDB 2C2L) reveals that only one m2 patch is exposed for a possible interaction with a flat membrane, the other being on the concave side of the dimer (*Figure 3a*). If docked to the surface involving m2, the TPR domain of the respective protomer would not be able to interact with a bulky chaperone any more (*Figure 3a*). This steric argument supports the assumption that CHIP association with membranes is prevented by bound HSP70 or HSP90.

## CHIP remains active on membranes

Since lipid binding can influence the activity of enzymes (*Fang et al., 2001*; *Leonard and Hurley, 2011*), we sought to assess the ubiquitylation function of CHIP upon interaction with membranes. None of the structural analyses revealed differences between soluble and liposome-associated protein implying full functionality of CHIP on membranes. First, CHIP remained dimeric and was still capable of oligomerization (*Figure 4a*). This is an important finding since monomerization of CHIP by the steroid detergent deoxycholate (*Figure 4—figure supplement 1a*) was shown to reversibly inactivate its ubiquitin ligase activity (*Nikolay et al., 2004*). The higher amount of oligomers from liposome samples does not necessarily mean that oligomerization is stronger on the liposomes. Because of the experimental constraints (due to dilution of non-bound protein in the supernatant), the input CHIP for the crosslinking reaction is less concentrated and thus cannot be compared quantitatively to the liposome-bound samples. The amount of secondary structure elements did not change upon binding to liposomes (*Figure 4b*), although CHIP is known to be a rather labile protein (*Graf et al., 2010*). Consistently, the susceptibility to protease hydrolysis on liposomes did not change (*Figure 4—figure supplement 1b*).

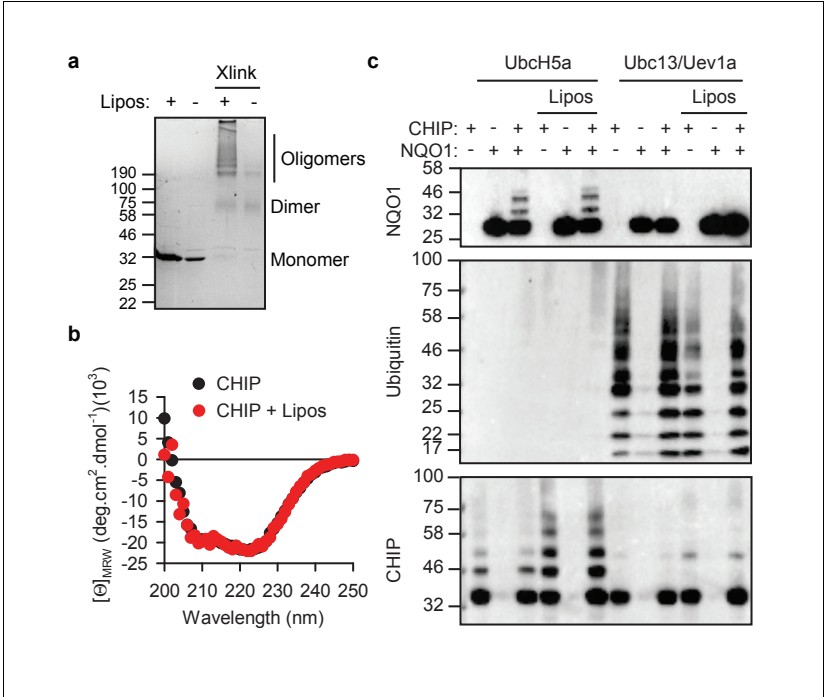

**Figure 4.** CHIP retains its activity on liposomes. (a) CHIP can oligomerize on PA-containing liposomes (Lipos) as determined by chemical crosslinking. Xlink, crosslinked samples. (b) CHIP does not change its global content of secondary structures upon binding to liposomes as determined by CD spectroscopy. Mean values (from three independent experiments) at respective wavelength are plotted. (c) While attached to liposomes, CHIP remains capable to ubiquitylate the substrate protein NQO1 and form unattached K63-conjugated chains. *In vitro* ubiquitylation assays were performed using as E2 either UbcH5a (for substrate and auto-ubiquitylation) or the heterodimer Ubc13/Uev1a (for unattached ubiquitin K63 conjugation). One representative out of three independent experiments is shown.

DOI: https://doi.org/10.7554/eLife.29388.008

The following figure supplement is available for figure 4:

**Figure supplement 1.** Characterization of CHIP associated with liposomes.
DOI: https://doi.org/10.7554/eLife.29388.009

As an E3 ubiquitin ligase, CHIP collaborates with several E2 ubiquitin-conjugating enzymes providing similar binding surface for interaction (*Xu et al., 2008*; *Zhang et al., 2005*). We used UbcH5a and Ubc13/Uev1a to compare soluble and membrane-bound CHIP regarding its K48- and K63-ubiquitylation activity, respectively (*Figure 4c*). NAD(P)H:quinone oxidoreductase 1 (NQO1) P187S mutant served as a substrate protein because it is known to be ubiquitylated by CHIP with and without assistance of chaperones (*Martínez-Limón et al., 2016*; *Tsvetkov et al., 2011*). Auto-ubiquitylation of CHIP is supported by UbcH5a as well (*Zhang et al., 2005*) and seemed to proceed somewhat stronger on liposomes (*Figure 4c*). K63-based ubiquitin polymerization by CHIP does not require a substrate protein (*Zhang et al., 2005*) and, contrary to auto-ubiquitylation, was slightly weaker on liposomes (*Figure 4c*). Ubiquitin K63-linking activity is especially interesting because it is known to regulate a number of membrane-related processes, such as receptor internalization, multivesicular body formation and autophagy (*Erpapazoglou et al., 2014*).

In summary, since no qualitative differences could be detected comparing the soluble and liposome-bound ligase, we concluded that CHIP on membranes remains functional and is capable of ubiquitylating the newly encountered subcellular proteome and produce free K63-linked ubiquitin chains.

## CHIP reorganizes the cellular proteome and architecture

Presuming activity of membrane-bound CHIP, we finally wanted to get insight into its cellular functions by identifying interactors of and proteome changes caused by chaperone-free CHIP. To avoid

dimerization of the transfected K30A variant with the endogenous wild-type CHIP, CHIP-deficient mouse embryonic fibroblasts were used in the subsequent experiments (*Figure 5—figure supplement 1a*). CHIP was implicated in HSF1 activation during stress response (*Dai et al., 2003*). However, early translocation of HSF1 into nucleus during heat shock was not affected by the absence of CHIP (*Figure 5—figure supplement 1b*). For comparison, we sought conditions when CHIP-K30A loses its membrane localization. One of the cellular pathways that generates CHIP-interacting phosphatidic acid (PA) is the hydrolysis of phosphatidylcholine by members of the phospholipase D (PLD) family (*Foster et al., 2014*). The PLD isoform-unspecific inhibitor FIPI and the inhibitor of PLD1 (VU0155069), but not that of PLD2 (CAY10594), efficiently released chaperone-free CHIP into the cytosol (*Figure 5a*). This was not due to changes in the abundance of CHIP (*Figure 5—figure supplement 1c*) or general disturbance of membrane structure as controlled by the correct localization of farnesylated EGFP (*Figure 5—figure supplement 1d*). From the other side, it was also possible to release chaperone-free CHIP from the plasma membrane using inhibitors of PI4P-generating type III phosphoinositol-4 kinases (PI4KIII) (*Figure 5—figure supplement 1e,f*). Wortmannin is a specific inhibitor of PI3K, however, it inhibits PI4KIII if used at higher concentrations (*Nakanishi et al., 1995*). Mass spectrometry analysis of EGFP-CHIP-K30A pull-downs from transiently transfected fibroblasts in the absence and presence of a PLD1 inhibitor was performed (*Figure 5—source data 1*). The comparison of CHIP over background enrichment allowed high-confidence identification of a set of chaperone-free CHIP interactors (*Figure 5b*, *Figure 5—source data 2*). Chaperone-independent ubiquitylation by CHIP has been sporadically reported in several instances; however, to our knowledge, this is the first high-throughput *in vivo* identification of CHIP interactors/substrates not associated via TPR-bound HSP70 or HSP90. Interestingly, more proteins interacted with chaperone-free CHIP when it was released into the cytosol (*Figure 5b*) suggesting pronounced 'stickiness' of the mislocalized ligase. The overlap between substrate sets was minimal (*Figure 5c*), which can be considered as biochemical verification of altered localization upon PLD1 inhibition. Most of the interactors reside at the plasma membrane or at one of the organelles confirming the broad potential of chaperone-free CHIP to modify cellular architecture (*Figure 5d*).

To set up conditions that avoid the confounding side-effects of heat shock yet freeing wild-type CHIP from chaperones, we transiently overexpressed the untagged ligase (together with low amounts of EGFP for the identification of the transfectants) in CHIP-deficient murine embryonic fibroblasts. We chose to use wild-type CHIP instead of K30A variant here since the interaction of the latter with E2 ubiquitin-conjugating enzymes was reported to be affected (*Narayan et al., 2015*). As a control, EGFP-only transfected fibroblasts were used. 24 hr later, the cells were sorted to enrich the transfected cells from one third to an almost pure population (*Figure 5—figure supplement 1g*) and proteome changes were measured by means of label-free quantitative mass spectrometry (*Figure 5—source data 3*). Correlation of the measured intensities from four independent experiments revealed robust repeatability resulting in a highly overlapping set of measured proteins (*Figure 5—figure supplement 1h*) and allowing identification of 56 significant changes (*Figure 5e*, *Figure 5—source data 4*). There were more proteins with a reduced rather than an increased abundance, 37 versus 19, respectively, which is an expected bias considering the involvement of CHIP in protein degradation. The accumulation of at least some of the proteins might be a consequence of the recently discovered noncanonical ubiquitylation by CHIP (*Ronnebaum et al., 2013*); however, indirect degradation loops leading to the abundance changes are conceivable as well.

Golgi-specific phosphatidylinositol-4-phosphate (*Stahelin et al., 2014*) showed specificity to CHIP in lipid array experiments (*Figure 1f*). GO analysis of CHIP interactors scored 'Golgi organization' GOBPslim term (biological processes) as the second-highest enrichment among significantly changed proteins (*Figure 5f*). This is especially interesting because the Golgi apparatus is known to fragment under heat shock (*Petrosyan and Cheng, 2014*; *Welch and Suhan, 1985*). Coexpression of the β−1,4-galactosyltransferase1-EGFP marker (*Cole et al., 1996*) verified fragmentation of the Golgi apparatus following CHIP overexpression (*Figure 5g*). In a complementary setup, CHIP-deficient fibroblasts failed to increase fragmentation of the Golgi apparatus during heat shock (*Figure 5h*) further supporting the involvement of CHIP in the reorganization of membrane-bound organelles during proteostasis stress. In addition to the changes in the 'Golgi organization' group (*Figure 5—figure supplement 1i*), several other components related to the Golgi apparatus were affected by CHIP overexpression as well. Some proteins showed lower abundance, for example, Golgi integral membrane protein 4 (Golim4) was reduced to 62% of its normal levels; others were

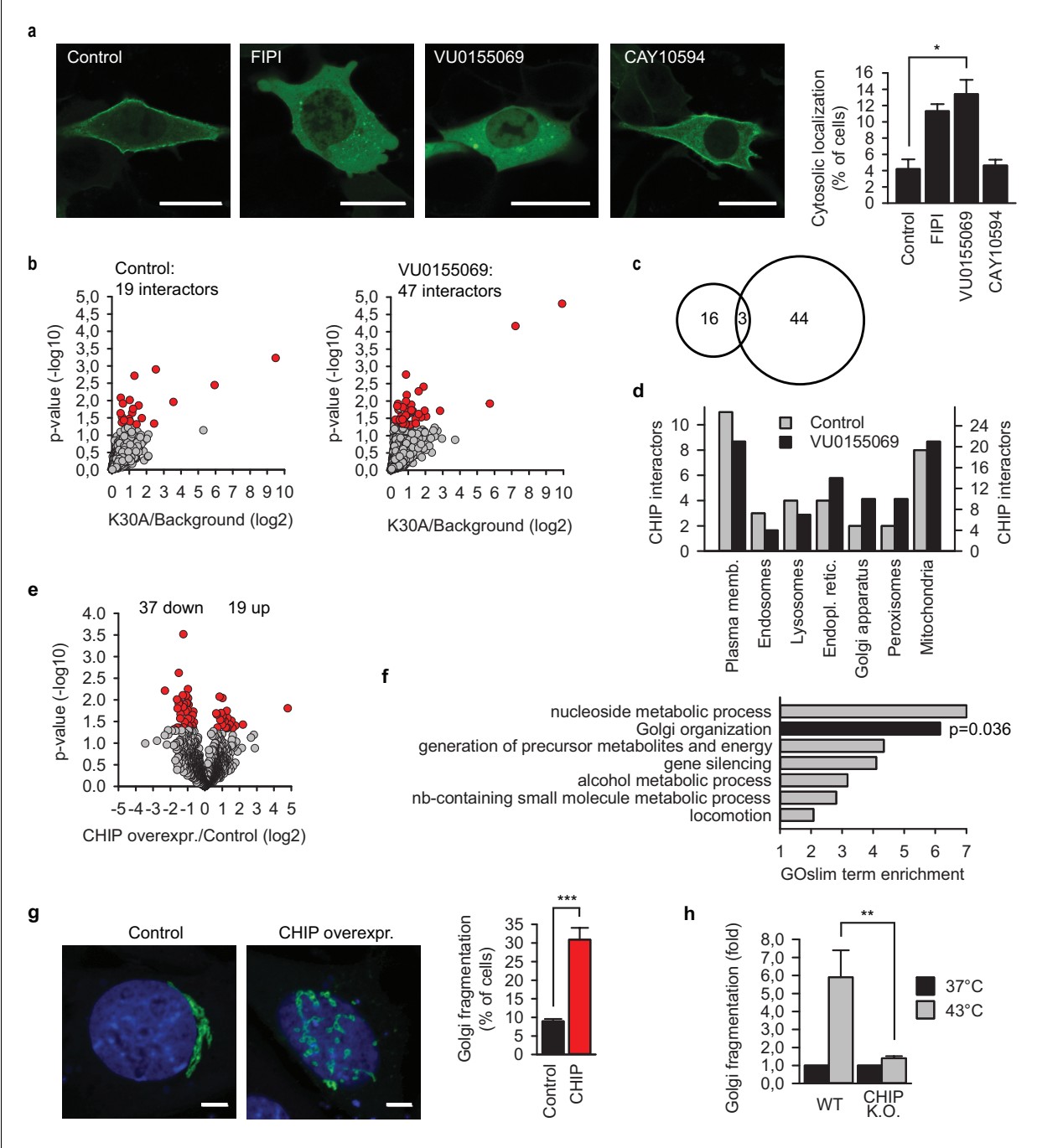

**Figure 5.** CHIP reorganizes cellular proteome and architecture. (**a**) Inhibition of phospholipase D (PLD) releases EGFP-CHIP-K30A from cellular membranes. FIPI, isoform independent inhibitor of PLD; VU0155069 and CAY10594, inhibitors of PLD1 and PLD2, respectively. Scale bar 20 mm. *p<0.05, chi-square analysis; N = 4 independent experiments (mean ± SD). (**b**) Interactors of EGFP-CHIP-K30A in transiently transfected murine embryonic fibroblasts (MEF). Phospholipase D was inhibited using 500 nM VU0155069. EGFP pulldowns were prepared as detailed in Methods and label-free mass spectrometry was used to quantitate proteins associated with CHIP. N = 5 biologically independent experiments. Volcano plots indicate average enrichment of individual protein levels. Red color marks significantly enriched proteins as determined by two sample t-test (p-value<0.05) (**c**) Minimal overlap of CHIP interactors from control and upon phospholipase D inhibition visualized by a venn diagram. (**d**) Subcellular localization of CHIP interactors as assigned by GeneCards database (www.genecards.org). According to the database, proteins may display multiple localizations. The scales of y-axis were adjusted according to the interactome sizes (19 versus 47). (**e**) Label-free quantitative mass spectrometry was used to determine proteome changes in MEFs upon overexpression of CHIP for 24 hr (N = 4). Volcano plot indicates average changes of individual protein levels. Red color marks significantly changed proteins as determined by two sample t-test (p-value<0.05). (**f**) GOBPslim terms enriched above two in the group of proteins significantly changed upon CHIP overexpression. Fisher exact test was used to determine statistical significance. (**g**) Morphology of the Golgi

*Figure 5 continued on next page*

*Figure 5 continued*

apparatus upon transient CHIP overexpression in MEFs assessed by co-transfected Golgi marker. DAPI stain in blue. Scale bar 5 mm. ***p<0.001, chi-square analysis; N = 3 independent experiments (mean ± SD). (h) Fragmentation of Golgi apparatus during heat shock in wild-type (WT) and CHIP knock-out MEFs (CHIP K.O.). Fraction of cells with fragmented Golgi at 37°C was set to 1. Increase of cells with fragmented Golgi after 30 min at 43°C is plotted as mean ± SD from three independent experiments. **p<0.01, t-test analysis.

DOI: https://doi.org/10.7554/eLife.29388.010

The following source data and figure supplement are available for figure 5:

**Source data 1.** LFQ of EGFP-CHIP-K30A interactors under the specified conditions.
DOI: https://doi.org/10.7554/eLife.29388.012
**Source data 2.** Significantly enriched proteins in EGFP-CHIP-K30A pull-downs.
DOI: https://doi.org/10.7554/eLife.29388.013
**Source data 3.** LFQ of proteome changes under the specified conditions.
DOI: https://doi.org/10.7554/eLife.29388.014
**Source data 4.** Significant proteome changes upon CHIP overexpression in murine embryonic fibroblasts.
DOI: https://doi.org/10.7554/eLife.29388.015
**Figure supplement 1.** Cellular reorganization by chaperone-free CHIP.
DOI: https://doi.org/10.7554/eLife.29388.011

upregulated, for example, the amount of Rab GDP dissociation inhibitor Gdi2 doubled. It will be interesting to uncover the exact mechanisms and functional consequences of Golgi fragmentation by chaperone-free CHIP (effector function). The central role of chaperone availability in this mechanism is supported by the observation that the overexpression of HSP70 inhibited fragmentation of the Golgi apparatus by CHIP (*Figure 5—figure supplement 1j*).

## Discussion

Early on, CHIP was shown to participate in the quality control and degradation of membrane proteins. The best-known example among its substrates is the ΔF508 mutant of chloride channel CFTR (*Meacham et al., 2001*). Many molecular details regarding the recognition and ubiquitylation of membrane proteins by CHIP *in vivo* and *in vitro* have been elucidated since (*Matsumura et al., 2013*; *Younger et al., 2006*). The unifying aspect of the analyzed processes has been the recognition of substrates by CHIP-bound molecular chaperones HSP70 and HSP90 (*Arndt et al., 2007*). In addition to the endoplasmic reticulum-related quality control, CHIP has been shown to participate in the quality control of plasma membrane-resident proteins (*Okiyoneda et al., 2010*). This activity also entails the participation of HSP70 and HSP90 (*Okiyoneda et al., 2010*). In contrast, here we describe the function of CHIP which relies on the chaperone-independent recruitment of the ubiquitin ligase to membrane-bound subcellular compartments. In fact, association with a chaperone would suppress this pathway of CHIP recruitment to membranes because its TPR domain is responsible for the mutually exclusive binding to either chaperones or membranes. It remains to be clarified how an additional determinant of the phospholipid specificity, the m2 patch, contributes to the TPR domain-driven association of CHIP with membranes. An electrostatic attraction of the positively charged cluster to the negatively charged membrane surfaces can be hypothesized, but remains to be tested. Indeed, a similar positively charged patch in vicinity of m2 does not influence CHIP interaction with lipids (*Figure 3*). At this stage, an allosteric effect upon the TPR domain cannot be excluded, especially because of the high flexibility of CHIP in general and of the m2 patch in particular (*Graf et al., 2010*).

Membrane specificity of chaperone-free CHIP is key to understanding CHIP-assisted reorganization of the cellular architecture during severe conformational stress. We addressed this question by means of analytical *in vitro* approaches which revealed the preference of the ligase for phosphatidic acid (PA) and phosphatidylinositol monophosphates. PA is interesting in the context of stress response for several reasons. First, in plants PA is known to be involved in response to many diverse stressors, such as high salt, drought, cold, aluminium exposure, to mention a few (*Liu et al., 2013*). Second, the constitutive levels of PA in membranes are comparably high, yet they can change to functionally significant extents depending on metabolic or stress conditions (*Fang et al., 2001*). Third, the involvement of PA-generating enzymes from the phospholipase D family in diverse

pathologies has been established, advancing the development of therapeutics and thus creating possibilities to intervene (*Frohman, 2015*). In this regard, Alzheimer's disease is an interesting example, having been associated with different isoforms of phospholipase D. On the other hand, phosphatidylinositol monophosphates (PIPs) are less abundant, but highly relevant as tags of organelle identity. For example, PI3P is enriched on early endosomes and PI4P marks the Golgi apparatus and plasma membrane (*Hammond and Balla, 2015*). PI3P and PI5P are less abundant than PI4P (ca. ten and hundred times, respectively) (*Hammond and Balla, 2015*), thus their affinity to CHIP might be an evolutionary byproduct with little functional consequences. At this stage, the relative importance of PA versus PI4P for *in vivo* relocalization of CHIP during stress remains unclear.

Discussing cellular localization approaches, we would like to emphasize the highly dynamic nature of CHIP interaction with membranes. It is strikingly evidenced by the undocking of CHIP from membranes during paraformaldehyde fixation of the samples, a standard processing before fluorescence microscopy. Another challenge is the highly suprastoichiometric amounts of chaperones with respect to CHIP levels in the cytosol which prevents freeing of CHIP even under stress conditions.

Summarizing, we propose a simple mechanism of how mammalian cells can sense the catastrophic extent of protein misfolding and aggregation in the cytosol and how membraneous compartments are acutely modified in response. Under normal conditions, HSP70- and HSP90-interacting proteins, such as CHIP, are associated with molecular chaperones and remain in the cytosol because their compartment-targeting determinants are shielded by that association. Acute conformational stress leads to accumulation of misfolded species with a high affinity for chaperones that, in turn, bind to the fresh aggregates following the law of mass action. Previous interactors of chaperones are now freed from shielding, thereby exposing their compartment-specific determinants allowing them to depart to new destinations. There, depending on functional specialization, relocalized proteins turn into effectors and modify newly encountered sets of proteins. Specifically in the case of CHIP, this would involve docking onto phosphatidic acid- and phosphatidylinositol-4-phosphate-enriched cellular compartments and subsequent ubiquitylation of compartment-specific proteins. The small fraction of relocalized CHIP indicates that the PN in the given cell line is strong enough to handle the ensuing misfolding and aggregation, with most of CHIP remaining in the cytosol (*Figure 1a*).

Immediate reorganization and adaptation of organelles is a less understood aspect of cytosolic stress response compared to the transcriptional and post-transcriptional reactions, such as HSF1-driven activation of stress gene transcription (*Akerfelt et al., 2010*) or eIF2α-mediated attenuation of protein synthesis (*Schneider and Bertolotti, 2015*). Importantly, the final outcome of the organelle reorganization probably depends on the capacity of a given cell to upregulate compensatory amounts of molecular chaperones. For the sake of the whole organism, metazoans might need to identify and sacrifice those cells which have failed to do so. An altruistic self-sacrifice involving membrane-bound organelles can be envisioned as an extension of the mechanism we propose. Although reversible under limited heat shock, Golgi fragmentation is a well-known marker or even the cause of cellular toxicity upon different stressors, including protein aggregation (*Alvarez-Miranda et al., 2015*; *Haase and Rabouille, 2015*).

Dissociation of CHIP from molecular chaperones under conformational stress resembles activation of HSF1 when the liberated protein trimerizes and translocates to the site of its transcriptional activity (*Morimoto, 1993*). However, molecular determinants of the inhibitory interaction between chaperones and HSF1 are less clear compared to the TPR-based mechanism described here. Involvement of the TPR domain poses a question regarding the generality of the mechanism. In addition to CHIP, a number of other members from the PN, such as HOP, DnaJC7, FKBP51, FKBP52 among others, engage their TPR domains to interact with HSP70 and HSP90 (*Assimon et al., 2015*). The organelle adaptation described here, if used by other co-chaperones with distinct functionality, would diversify the response to acute stress. Finally, unmasking of the determinants of cellular localization must not be restricted to protein misfolding. For example, C-terminal phosphorylation of HSP70 and HSP90, such as that which takes place during a proliferative boost, was shown to critically affect TPR domain-based associations (*Muller et al., 2013*). Thus, comprehension of the scope and details of the subcellular reorganization of protein quality control machinery bears wide biological implications.

# Materials and methods

## Reagents, plasmids and antibodies

Phospholipase D inhibitors FIPI, VU0155069 and CAY10594 were purchased from Caymen Chemicals (Ann Arbor, MI), wortmannin from Enzo (Farmingdale, NY), PI4KIIIβ inhibitor IN-10 from MedChem Express (Monmouth Junction, NJ), HSP70 inhibitor VER-155008 from Sigma-Aldrich (St. Louis, MO), HSP90 inhibitor 17-(Allylamino)−17-demethoxygeldanamycin (17-AAG) from Biomol (Germany). 1,2-dioleoyl-$sn$-glycero-3-phosphocholine (DOPC) and 1,2-dipalmitoyl-$sn$-glycero-3-phosphate (phosphatidic acid, PA) were from Avanti Polar Lipids (Alabaster, AL), 1,2-dipalmitoyl-$sn$-glycero-3-phosphoinositol-4-phosphate (phosphadtidylinositol-4-phosphate, PI4P) was from Echelon Bioscience (Salt Lake City, UT). All other chemicals were from Sigma-Aldrich if not indicated otherwise. Octapeptide GPTIEEVD was custom-synthetized by GenScript (Piscataway, NJ).

For mammalian expression of EGFP, the expression vector pEGFP-N1 from Clontech (Mountain View, CA) was used. N-terminal EGFP-CHIP fusion protein was constructed by cloning human CHIP coding sequence in pEGFP-C2 vector (Clontech) using HindIII and BamHI restriction sites. A fusion between the N-terminalβ−1,4-galactosyltransferase1 and EGFP was used as a marker of the Golgi apparatus, farnesylated EGFP as a marker of plasma membrane (both constructs from Clontech). Untagged and 3xFLAG-tagged human CHIP for mammalian expression were constructed in pCMV-10 vector (Sigma-Aldrich). HSP70 was amplified from human cDNA and cloned into pcDNA3. 1 expression vector without a tag. HSP70 lacking the last amino acid (HSP70ΔD) was prepared by site-directed mutagenesis. Vectors of full-length CHIP, UbcH5a and mutant NQO1 for bacterial expression were prepared as described (*Martínez-Limón et al., 2016*). The TPR domain bacterial expression vector (TPR) was created by exchanging glutamate152 codon to a stop codon in the full-length CHIP vector. CHIP without TPR domain (ΔTPR) was cloned by deleting the sequence between serine25 and argininge130 by means of PCR. To construct the bacterial expression vector HSP70-C, the sequence of human HSPA1A coding last 260 amino acids of the protein was cloned into pPROEX HT (Invitrogen, Waltham, MA). Site-directed mutagenesis was used to create K to A (30 aa), KKK to AAA (143–145 aa) and KRKK to AAAA (221–224 aa) mutants of CHIP in respective vectors for mammalian and bacterial expression.

Antibody against ubiquitin (P4D1) was from Santa Cruz Biotechnology (Dallas, TX), antibodies against eIF4E (9742), HSF1 (D3L8I) lamin B1 (D9V6H) and GAPDH (14C10) were from Cell Signaling (Danvers, MA). Anti-C-CHIP (C9243), anti-N-CHIP (C9118) and anti-NQO1 (N5288) antibodies were from Sigma-Aldrich. Anti-HSC70 (1B5) was from Enzo.

## Cell culture and transfection

Immortalized murine embryonic fibroblasts (MEF) were gift from Dr. A. Reichert (Düsseldorf University). The identity of the cells has been authenticated by analyzing highly polymorphic short tandem repeat loci (Microsynth, Switzerland). Cells were cultured in Dulbecco's modified Eagle's medium (DMEM) supplemented with 10% fetal bovine serum (FBS), 2 mM L-glutamine, 100 IU/ml penicillin G, 100 µg/ml streptomycin sulphate and non-essential amino acids (Gibco, Waltham, MA) at 37°C and 5% $CO_2$ in a humidified incubator. MEF clones lacking CHIP (ΔCHIP) were engineered using CRISPR/Cas9 system (*Ran et al., 2013*). pSpCas9n(BB)−2A-GFP (PX461) and pSpCas9n(BB)−2A-Puro (PX462) plasmids were a gift from Feng Zhang (Addgene #48140 and # 48141). Cell lines were free of Mycoplasma contamination according to the PCR Mycoplasma test kit II (AppliChem, Germany).

For microscopy of CHIP localization, $4 \times 10^6$ MEF cells were transfected by means of electroporation with 10 µg EGFP-CHIP or EGFP-CHIP-K30A. Transfected cells were seeded on poly-lysine-coated cover slides in a 12-well plate. 6 hr after transfection medium was exchanged. 24 hr after transfection cells were processed for microscopy without fixation as specified below.

For phospholipase D inhibitor experiments, $2 \times 10^5$ MEF cells per well were seeded on poly-lysine-coated cover slides in a 12-well plate. Transfection was carried out 10 hr later using 1 mg/ml polyethylenimin (PEI) solution and 1 µg of EGFP-CHIP-K30A (9:1 PEI to DNA ratio). Medium was exchanged 12 hr after transfection. 24 hr after transfection PLD inhibitors or DMSO were added in serum-free DMEM. 12 hr after inhibitor treatment slides were processed for microscopy without fixation as specified below.

For live imaging during heat stress, $0.1 \times 10^6$ MEF cells per well were seeded in a 24-well Cell Imaging Plate with clear film bottom (Eppendorf, Germany). Cells were transfected with 1 µg EGFP-CHIP or EGFP as control (6:1 PEI to DNA ratio).

For analysis of CHIP relocalization to the plasma membrane upon HSP70 and HSP90 inhibition, $2 \times 10^5$ MEF cells per well were seeded in a 12-well plate on poly-lysine coated cover slides. Cells were transfected with 1 µg EGFP-CHIP or EGFP as control (6:1 PEI to DNA ratio). 20 µM VER-155008 or 20 µM 17-AAG were added in serum-free medium 24 hr later and incubated for 4 hr before mounting for microscopy without fixation.

For MS analysis of chaperone-free CHIP interactors, $4 \times 10^6$ ΔCHIP MEF were seeded on 10 cm dishes. Transfection was carried out 10 hr later with 10 µg EGFP-CHIP-K30A or empty vector (6:1 PEI to DNA ratio). Medium was refreshed 12 hr past transfection. For treatment with PLD inhibitors, the medium was exchanged 24 hr past transfection to DMEM with no FBS and containing 500 nM PLD inhibitor (VU0155069) or DMSO. 12 hr later cells were washed with PBS and harvested for lysis.

For label-free mass spectrometry analysis, two times $8 \times 10^6$ ΔCHIP MEF (clone 22) were each co-transfected with 3 µg EGFP and 30 µg CHIP or 30 µg control vector by electroporation. Transfected cells were seeded on two 10 cm dishes and medium was exchanged 6 hr later. 24 hr after transfection, cells were trypsinized, pooled to a concentration of $10 \times 10^6$ cells/ml and sorted using S3 cell sorter (Bio-Rad Laboratories, Hercules, CA).

For analysis of Golgi morphology, $2 \times 10^5$ ΔCHIP MEF (clone 22) per well were seeded on poly-lysine-coated cover slides in a 12-well plate. At 70% confluence and 24 hr after seeding, medium was exchanged and cells were transfected with a 1 mg/ml PEI solution using 200 ng β−1,4-galacto-syltransferase1-EGFP and 800 ng CHIP (9:1 PEI to DNA ratio). Controls were transfected with an equivalent amount of empty plasmid. 36 hr past transfection cells were fixed and processed for microscopy as described below.

For analysis of Golgi morphology under heat stress, $2 \times 10^5$ ΔCHIP and WT MEF were seeded on poly-lysine-coated cover slides in a 12-well plate. At 70% confluence and 24 hr past seeding, medium was refreshed and cells were transfected with a 1 mg/ml PEI solution (9:1 PEI to DNA ratio) using 1 µg β−1,4-galactosyltransferase1-EGFP (Clontech).

## Immunoprecipitation and western blotting

For immunoprecipitation, $8 \times 10^6$ MEF were electroporated at 300V and 950 µF in 400 µl intracellular buffer/25% FBS using 20 µg 3xFLAG-CHIP or empty vector (40 µg total DNA). Cells were washed and plated on 10-cm tissue culture dishes in 10% FBS/DMEM; 6 hr later medium was refreshed. 24 hr after transfection, medium was supplemented with 25 mM HEPES NaOH pH 7.4, cells were exposed to 43°C for 60 min, washed twice with PBS, scraped from the plates and lysed in 450 µl lysis buffer (20 mM HEPES KOH pH 7.4, 100 mM KCl, 10 mM MgCl$_2$, 20 mM sodium molibdate, 10% (w/v) glycerol, 0.5% (v/v) IGEPAL) with protease inhibitors. Lysates were pulse-sonicated to shear DNA, incubated on ice for 15 min, centrifuged at 10,000 g for 10 min and normalized in regard to protein concentration. Immunoprecipitation was performed using 1 µl M2 affinity gel (Sigma) and 9 µl Protein G sepharose (GE Healthcare, Chicago, IL) for 2 hr at 4°C. The precipitates on mini-cartridges were washed three times with lysis buffer and eluted with 40 µl sample buffer at 95°C for 3 min. β-mercaptoethanol was added to the eluates which were then analyzed by SDS-PAGE and western blotting using anti-FLAG and anti-HSC70 antibodies. Chemiluminescence images were acquired with the ChemiDoc MP imaging system and bands quantified using Image Lab 5.0 software (both from Bio-Rad Laboratories).

## Fluorescence microscopy

Zeiss LSM 780 inverted confocal microscope with a 63x oil immersion objective was used.

For microscopy analysis of CHIP localization, samples were not fixed. Cover slides with living cells were mounted in culture medium on glass slides with parafilm spacers. Cover slides were sealed with vaseline and immediately imaged at RT. For live imaging during heat stress, the plate was moved to the microscope incubator at 37°C and 5% CO$_2$. Cells were imaged 24 hr after transfection at the respective temperature. For analysis of Golgi morphology upon CHIP overexpression, cover-slides were washed twice with PBS before cells were fixed with 4% paraformaldehyde for 20 min at RT. Cells were washed three times with PBS and subsequently stained with 1 µg/ml DAPI for 3 min

at RT. After removing residual DAPI by washing three times with PBS, cells were mounted in PBS onto glass slides and sealed using nail polish. For quantification, fields of view were chosen randomly. For analysis of Golgi morphology during heat shock, cells were incubated at 43°C for 30 min, then fixed with 4% PFA and stained with DAPI.

Membrane localization and Golgi morphology were analyzed manually using ImageJ from following numbers of cells (average of at least three independent experiments): 92 (*Figure 1a*), 142 (*Figure 1d*), 90 (*Figure 1—figure supplement 1b*), 125 (*Figure 1—figure supplement 1c*), 149 (*Figure 1—figure supplement 1d*), 84 (*Figure 3c*), 130 (*Figure 5a*), 206 (*Figure 5g*), 240 (*Figure 5h*), 93 (*Figure 5—figure supplement 1d*), and 137 (*Figure 5—figure supplement 1j*). In case of *Figure 5—figure supplement 1e and f*, the same cells were followed upon addition of inhibitors: 47 and 79, respectively.

## Immunofluorescence

For staining of stress granules after arsenite treatment, cells on polylysine-coated cover slides were washed three times with PBS and fixed for 15 min with 4% PFA at RT. Following washing with PBS, cells were permeabilized with acetone for 5 min at −20°C. Remaining acetone was removed by three washing steps before addition of 1% BSA/PBS for 1 hr at RT. Anti-eIF4E antibody (Cell Signaling) was used in a 1:100 dilution in 1% BSA/PBS for 1 hr at RT. Cover-slides were washed three times with PBS before addition of anti-rabbit IgG Fab2 Alexa Fluor (R) 647 (Cell Signaling, 4414) at 1:1000 in 1% BSA/PBS and incubated for 1 hr at RT protected from light. Aspiration of secondary antibody was followed by three washing steps with PBS and DAPI staining.

## Subcellular fractionation and chemical crosslinking

Cells were cultured on a 10 cm dish to 80–90% confluence. $20 \times 10^6$ cells/ml suspension was lysed in subcellular fractionation buffer (SF: 250 mM sucrose, 20 mM HEPES KOH pH 7.5, 10 mM KCl, 1.5 mM MgCl$_2$, 1 mM EDTA, 1 mM EGTA, 1 mM DTT, protease inhibitor mix from Sigma). Cells were disrupted by passing thirty times through a 26 G needle. The suspension was cleared by centrifugation at 4°C and 720 *g* for 5 min and two times at 10,000 *g* for 5 min. The cytosolic fraction was prepared by ultracentrifugation of the supernatant at 100,000 *g* for 1 hr at 4°C in a TLA120.1 fixed angle rotor (Beckman, Indianapolis, IN). The transparent membrane pellet was washed once with 400 µl SF buffer and then ultracentrifuged at 100,000 *g* for 45 min at 4°C. The washed pellet (membrane fraction) was resuspended in 80 µl CHIP lysis buffer (20 mM HEPES KOH pH 7.5, 100 mM KCl, 10 mM MgCl$_2$, 10% (v/v) glycerol, 0.5% (v/v) IGEPAL). 20 µl of total lysate, cytosolic and membrane fractions were incubated with 800 µM HSP70 C-terminal peptide GPTIEEVD for 30 min at 37°C followed by crosslinking with 0.025% glutaraldehyde for 10 min at 30°C. Crosslinking reaction was quenched with 100 mM Tris HCl pH 7.5 and analyzed by western blotting with anti-CHIP antibody.

## Recombinant protein purification

Human recombinant full length CHIP and its truncated versions were purified from *Escherichia coli* BL21 cells using Glutathion Sepharose 4B (GE Healthcare). Sepharose-bound GST-CHIP was subjected to overnight treatment with PreScission protease and the released tag-free protein was collected. Size exclusion chromatography on a HiLoad Superdex 200 column (GE Healthcare) followed to ensure the purity and homogeneity of proteins. All CHIP variants were kept in HSP70 buffer (25 mM HEPES KOH pH 7.5, 150 mM KCl, 5 mM MgCl$_2$, 5% glycerol, 1 mM DTT).

Human C-terminal HSP70 6xHis-tagged N-terminally was purified from *Escherichia coli* BL21 cells using 1 ml HisTrap column (GE Healthcare). Bound protein was eluted by applying an imidazole gradient. Fractions containing the protein were subjected to size exclusion chromatography using HiLoad Superdex 200 in HSP70 buffer. His-NQO1, His-UbcH5a, His-Uev1a and His-Ubc13 were purified similarly except that PBS/1 mM DTT was used during size exclusion chromatography. Purity of recombinant protein was evaluated by SDS-PAGE and Coomassie BB G-250 (CBB) staining. Concentration of proteins was determined measuring absorbance at 280 nm and using ProtParam (ExPASy) to calculate respective extinction coefficients. Proteins were aliquoted, snap-frozen and stored at −80°C.

## Isothermal titration calorimetry

ITC was carried out using a Nano ITC (TA instruments). The calorimetry cell contained 10 μM of the respective protein in HSP70 buffer. The 500-μl titration syringe was loaded with 200 μM GPTIEEVD peptide. Titrations were carried out performing 25 injections of 8 μl every 3 min. Data were acquired at 350 rpm stirring speed and constant temperature at 37°C. Heats of dilution were determined by injecting peptide into buffer. Corrected data were fitted to independent binding model using Nano-Analize Software (TA Instruments, New Castle, DE).

## Liposome preparation

Liposomes were prepared by mixing 1,2-dioleoyl-sn-glycero-3-phosphocholine (DOPC) with indicated amount (wt/wt) of either 1,2-dipalmitoyl-sn-glycero-3-phosphate (phosphatidic acid) or (wt/wt) 1,2-dipalmitoyl-sn-glycero-3-phosphoinositol-4-phosphate (PI4P) in a 1.5 ml reaction tube (Eppendorf). Lipid mixtures were dried under a nitrogen stream and kept for 1 hr under vacuum. Liposomes were rehydrated in 1 ml filtered reconstitution buffer (25 mM HEPES KOH pH 7.5, 50 mM NaCl) to a concentration of 10 mM lipids. Liposomes were reconstituted by shaking at 2000 rpm for 1 hr at 50°C, followed by sonication for 1 hr at 50°C in a water bath. The liposome suspension was subjected to five freeze-thaw cycles in liquid nitrogen to promote unilamellar vesicle formation, aliquoted in 200 μl and pulse-sonicated for 8 s at 45% output with MS72 sonotrode (Bandelin) to create a population of homogenously sized liposomes as determined using NanoSight (Malvern Instruments). Sonicated liposomes were pooled, aliquoted again, snap-frozen and stored at −80°C.

## Liposome floatation assay

1 mM liposomes and 0.5 μM CHIP were incubated in 150 μl reconstitution buffer for 30 min at 37°C. The suspension was adjusted to 30% sucrose by adding 100 μl of a 75% (w/v) sucrose solution in reconstitution buffer. The resulting high-sucrose solution was transferred to a 500 μl ultracentrifugation tube and overlaid with 200 μl of a 25% (w/v) sucrose solution and 50 μl reconstitution buffer. The sample was centrifuged at 240,000 g in a TLA120.1 fixed angle rotor for 1 hr at 25°C. The bottom (250 μl), middle (150 μl) and top (50 μl) fractions were collected from bottom to top using a 100 μl Hamilton syringe. Resulting fractions were analyzed by SDS-PAGE and western blotting after loading dilution-corrected amount of respective fractions.

## Liposome pelleting assay

2 mM liposomes were mixed with 2 μM recombinant protein in 50 μl reconstitution buffer. The suspension was incubated shaking at 450 rpm for 30 min at 37°C. Liposomes with bound protein were sedimented by ultracentrifugation at 100,000 g for 30 min at 25°C. Liposome pellet was washed with 50 μl of reconstitution buffer and sedimented again at 100,000 g for 30 min at 25°C. The pellet was resuspended in 10 μl reconstitution buffer, resolved by SDS-PAGE and analyzed by CBB staining. Images were acquired with ChemiDoc MP imaging system and quantified using Image Lab 5.0 software (both from Bio-Rad Laboratories). For competition assays, indicated amounts of C-terminal HSP70 were preincubated with 2 μM CHIP for 20 min at 37°C prior to co-sedimentation.

## Lipid-binding assay

Membrane-immobilized lipids (Membrane lipid strips P-6002 and PIP strips P-6001) were purchased from Echelon Research Laboratories. The strips were incubated with blocking buffer (3% fatty acid free BSA, 0.1% (v/v) Tween-20 in PBS pH 7.5) for 1 hr at RT. 0.5 μg/ml of purified wild-type or mutant CHIP was added in blocking buffer and incubated for 1 hr at RT. Strips were washed and bound proteins were detected using anti-C-CHIP antibody. For competition experiments, ten times molar excess of C-terminal HSP70 was added to CHIP in blocking buffer.

## Chemical crosslinking *in vitro*

5 μM of recombinant CHIP were crosslinked with 0.025% glutaraldehyde for 10 min at 30°C unless indicated otherwise. Crosslinking reaction was quenched with 100 mM Tris HCl pH 7.5 for 10–15 min at RT. Where indicated, HSP70 peptide at 800 μM was pre-incubated at 30°C for 20 min before adding glutaraldehyde. Crosslinking on 20% PA-containing liposomes was performed after

centrifugation on the resuspended pellet. For CHIP monomerization, fresh 10% sodium deoxycholate stock solution was used. Crosslinks were analyzed with SDS-PAGE and CBB staining.

## CD spectroscopy

Far-UV CD spectra of 10 μM CHIP in solution or bound to 20% PA-containing liposomes were recorded on a Jasco J-810 spectropolarimeter at 37°C in 10 mM $KH_2PO_4$ pH 7.5. Prior to measurement, samples were pre-incubated for 30 min at 37°C. Control samples without liposomes were supplemented with an equivalent amount of reconstitution buffer. Three repeat scans were obtained for each sample and subtracted from buffer alone baseline. Data were collected in 1 mm path cell (Helma Analytics) from 250 nm to 200 nm in 1 nm steps at 50 nm/min.

## Protease sensitivity assay

CHIP pelleted with 20% PA-containing liposomes was subjected to 0.5 ng/μl trypsin treatment at 37°C. Hydrolysis was stopped by adding hot reducing SDS sample buffer and boiling for 5 min. Samples were analyzed by SDS-PAGE and CBB staining. For controls without liposomes, protein was used at concentration assuming 30% binding to liposomes. Images were acquired with ChemiDoc MP imaging system and quantified using Image Lab 5.0 software (both from Bio-Rad Laboratories).

## *In vitro* ubiqitylation

*In vitro* ubiquitylation was performed in 50 mM Tris HCl pH 7.5, 50 mM NaCl, 10 mM $MgCl_2$, 2 mM DTT and 2 mM ATP. Reactions contained 100 nM UBE1 (Boston Biochem, Cambridge, MA), 2 μM either UbcH5a or Uev1A/Ubc13, 5 μM CHIP and 100 mM ubiquitin from bovine erythrocytes (Sigma-Aldrich). 2.5 μM NQO1 P187S mutant was added as substrate where indicated. Reactions proceeded at 37°C for 1 hr and then were stopped by adding reducing SDS sample buffer and boiling samples at 95°C for 5 min. Protein modification was analyzed by SDS-PAGE and immunoblotting using antibodies against NQO1, CHIP and ubiquitin. Where indicated, CHIP pelleted with 20% PA-containing liposomes was used instead of soluble protein.

## Isolation of nuclei

To analyze the translocation of HSF1 into nucleus, the lysis gradient protocol (*Schatzlmaier et al., 2015*) with small modifications was used. Cells were seeded on two 10 mm plates at $5 \times 10^6$ cells/plate per condition and medium exchanged 6 hr later. Next day, lysis gradient containing 0.5% NP-40 and 0.5% *n*-dodecyl-β-d-maltoside (DDM) was prepared freshly before heat shock. Cells were heat shocked at 43°C for 60 min, washed with cold PBS and scraped into 700 μl cold 10% DMEM. Cell suspension ($7.5 \times 10^6$) was loaded onto lysis gradient in a 14 ml tube and centrifuged at 1000 *g* for 10 min at 4°C. Isolated nuclei from the lower interface were washed with 600 μl of nuclei isolation buffer (0.25 M sucrose, 10 mM Tris HCl pH 7.4, 25 mM KCl, 5 mM $MgCl_2$) at 1000 *g* for 10 min at 4°C twice. Washed nuclei were resuspended in 100 μl CHIP lysis buffer, incubated on ice for 15 min and then sonicate for 10 s at 52% output with MS72 sonotrode (Bandelin, Germany). Samples were normalized and the amount of HSF1 in nuclei was analyzed by western blotting.

## Mass spectrometry

### Sample preparation

#### Pulldowns

Cells were lysed in 300 μl lysis buffer (10 mM Tris HCl pH 7.5, 150 mM NaCl, 0.5 mM EDTA, 10 mM $Na_2MoO_4$, 0.5% NP-40) and DNA was sheared by sonication for 3 s. The lysate was cleared by centrifugation at 12,000 *g* for 15 min and protein amount was normalized to 1.25 μg/μl using dilution buffer (10 mM Tris HCl pH 7.5, 150 mM NaCl, 0.5 mM EDTA, 10 mM $Na_2MO_4$). The lysate was incubated with 20 μl of GFP-Trap-M magnetic beads (ChromoTek, Germany) for 90 min at 4°C. Beads were washed thrice with dilution buffer and twice with MS wash buffer (20 mM Tris HCl pH 7.5, 150 mM NaCl) and then snap-frozen in liquid nitrogen and stored at −80°C until further processing. Beads were re-suspended in 50 μl 8 M urea, 50 mM Tris HCl pH 8.5, reduced with 10 mM DTT for 30 min and alkylated with 40 mM chloroacetamide for 20 min at 22°C. Urea was diluted to a final concentration of 2 M with 25 mM Tris HCl pH 8.5, 10% acetonitrile and proteins were digested with trypsin/lysC mix (mass spec grade, Promega, Madison, WI) overnight at 24°C. Acidified peptides

(0.1% trifluoroacetic acid final concentration) were desalted and fractionated on combined C18/strong cation exchange StageTips. Peptides were dried and resolved in 1% acetonitrile, 0.1% formic acid.

### Proteomes

Cells were lysed in 150 µl lysis buffer (10% SDS, 150 mM NaCl, 100 mM Hepes NaOH pH 7.6) and DNA was sheared by sonication for 5 s. Cell lysates were cleared by centrifugation at 16,000 *g* for 3 min. 20 µg of total protein were diluted in 4% (w/v) SDS, 100 mM HEPES/NaOH pH 7.6, 150 mM NaCl, 0.1 M DTT and heated at 95℃ for 5 min. The lysates were then mixed with 200 µl 8 M urea, 50 mM Tris/HCl, pH 8.5 and loaded onto spin filters with 30 kDa cut-off (Microcon, Merck, Germany). The filter-aided sample preparation protocol (FASP) was essentially followed (*Wiśniewski et al., 2009*). Proteins were digested overnight with trypsin (sequencing grade, Promega). According to *Kulak et al. (2014)*, acidified peptides (0.1% trifluoroacetic acid final concentration) were desalted with C18 StageTips and fractionated with strong cation exchange (SCX) StageTips. The C18 trans-elution fraction was combined with the first of 6 SCX fractions. Peptides were dried and resolved in 1% acetonitrile, 0.1% formic acid.

### LC-MS/MS

LC-MS/MS was performed on a Q Exactive Plus equipped with an ultra-high pressure liquid chromatography unit Easy-nLC1000 and a Nanospray Flex Ion-Source (all three from Thermo Fisher Scientific, Waltham, MA). Peptides were separated on an in-house packed column (100 µm inner diameter, 30 cm length, 2.4 µm Reprosil-Pur C18 resin [Dr. Maisch GmbH, Germany]) using a gradient from mobile phase A (4% acetonitrile, 0.1% formic acid) to 30% mobile phase B (80% acetonitrile, 0.1% formic acid) for 60 min followed by a second step to 60% B for 30 min, with a flow rate of 250 nl/min. MS data were recorded in data-dependent mode selecting the 10 most abundant precursor ions for HCD fragmentation. The full MS scan range was set from 200 to 2000 m/z with a resolution of 70,000. Ions with a charge $\geq 2$ were selected for MS/MS scan with a resolution of 17,500 and an isolation window of 2 m/z. Dynamic exclusion of selected ions was set to 30 s. Data were acquired using the Xcalibur software (Thermo Fisher Scientific).

### Data analysis

MS raw files were analyzed with Max Quant (version 1.5.3.30) (*Cox and Mann, 2008*). The spectra were searched against the UniProtKB mouse FASTA database (downloaded in January 2016) for protein identification with a false discovery rate of 1%. Unidentified features were matched between runs in a time window of 2 min. Hits in three categories (false positives, only identified by site, and known contaminants) were excluded from further analysis. For label-free quantification (LFQ), the minimum ratio count was set to 1. Bioinformatic data analysis was performed using Perseus (version 1.5.2.6) (*Tyanova et al., 2016*). Statistical analysis between different conditions was performed on logarithmized LFQ intensities for those proteins that were found to be quantified in at least four out of five biological replicates (for pulldowns) or in at least three out of four biological replicates (for proteomes). Three significantly changed proteins showing high fluctuation (standard deviation higher than the respective average value) were excluded from in proteome analysis.

## Structure analysis

CHIP sequences from different organisms were compared using the multiple alignment tool T-coffee (*Notredame et al., 2000*) and the alignment was visualized with ESPript 3 (*Robert and Gouet, 2014*). UCSF Chimera (*Pettersen et al., 2004*) was used to analyze the predicted lipid-binding sites on CHIP (PDB 2c2l).

## Statistical analysis

All repetitions in this study were independent biological repetitions performed at least three times if not specified differently. To identify significantly changed proteins, a two-sample t-test analysis of grouped biological replicates was performed using a p-value cutoff of 0.05. Categorical annotation was added in Perseus and a Fisher exact test with a p-value threshold of 0.05 was run for GOslim term enrichment analysis. Statistical significance for categoric readouts in microscopy (membrane

versus cytosol localization and Golgi apparatus fragmentation) were analyzed by chi-square analysis. Means and standard deviations were calculated from at least three independent experiments.

## Acknowledgements

We thank members of Protein aggregation group for help and discussions. We thank Dr. Margot Scheffer for critical reading of the manuscript. This work was supported by the European Research Council grant 311522-MetaMeta (YK, TBS, GC and RMV) the Cluster of Excellence 'Macromolecular Complexes' EXC 115 (W-HL and RMV), LOEWE grant Ub-Net (AML), the Austrian Science Fund (FWF) Erwin Schrödinger Fellowship J3987-B21 (HFH), and the Emmy Noether Program of the German Research Foundation EN608/2-1 (RE and HFH).

## Additional information

### Funding

| Funder | Grant reference number | Author |
|---|---|---|
| European Research Council | StG-311522 | Yannick Kopp<br>Tobias B Schuster<br>Giulia Calloni<br>R Martin Vabulas |
| Deutsche Forschungsgemeinschaft | EXC115 | Wei-Han Lang<br>R Martin Vabulas |
| LOEWE | Ub-Net | Adrián Martínez-Limón |
| Deutsche Forschungsgemeinschaft | Emmy Noethe Program EN608/2-1 | Harald F Hofbauer<br>Robert Ernst |
| Austrian Science Fund | Erwin Schrödinger Fellowship J3987-B21 | Harald F Hofbauer |

The funders had no role in study design, data collection and interpretation, or the decision to submit the work for publication.

### Author contributions

Yannick Kopp, Wei-Han Lang, Tobias B Schuster, Adrián Martínez-Limón, Harald F Hofbauer, Robert Ernst, Giulia Calloni, Investigation; R Martin Vabulas, Conceptualization, Supervision, Writing—original draft, Writing—review and editing

### Author ORCIDs

Harald F Hofbauer (iD) https://orcid.org/0000-0003-2617-5901
R Martin Vabulas (iD) http://orcid.org/0000-0001-8519-4489

### Decision letter and Author response

Decision letter https://doi.org/10.7554/eLife.29388.017
Author response https://doi.org/10.7554/eLife.29388.018

## Additional files

### Supplementary files

• Transparent reporting form
DOI: https://doi.org/10.7554/eLife.29388.016

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
