## [Decision Letter]

Thank you for submitting your article "CHIP as a membrane-shuttling proteostasis sensor" for consideration by *eLife*. Your article has been reviewed by three peer reviewers, one of whom, Davis Ng is a member of our Board of Reviewing Editors and the evaluation has been overseen by Anna Akhmanova as the Senior Editor. The following individuals involved in review of your submission have agreed to reveal their identity: Vytas A Bankaitis (Reviewer #3).

The reviewers have discussed the reviews with one another and the Reviewing Editor has drafted this decision to help you prepare a revised submission.

Summary:

The authors propose the intriguing hypothesis that the Hsp70 associated E3 enzyme CHIP associates with specific membrane lipids to act as a "proteostasis sensor". The data supporting this role consist of intracellular localization of CHIP-EGFP under varying conditions, cell fractionation experiments, filter binding assays of CHIP to specific lipids, liposome binding experiments, and CHIP pull-down experiments coupled with MS. In general, the data are convincing that Hsp70 unassociated CHIP can associate with lipids and this association is related to membrane fragmentation caused by stress. The reviewers are in agreement that the study is interesting, breaks new ground, and appropriate for publication in *eLife*. However, the reviewers also agree that substantial gaps remain that can be ameliorated with some additional experimentation. Of particular concern is the functional relationship between CHIP relocalization and Golgi fragmentation. The reviewers also suggest another round of text editing (perhaps with assistance of colleague not directly involved in the work) to improve the readability and clarity of the manuscript.

Essential revisions:

1) It would be useful to quantify plasma membrane association for images other than the heat shock. For example, quantification of plasma membrane association of CHIP for images as shown in Figure 1 and Figure 1—figure supplement 1. Considering the subtlety of some of this phenotype (only ~20% of cells for the quantified heat shock), this is important to demonstrate the indicated effects.

2) It would be interesting to see how universal this response is to other types of cytosolic stress. For example, As(III) induces protein misfolding in the cytosol and activation of the HSR. It would be interesting to see if As(III) induces a similar response.

3) It's unclear how the CHIP membrane association phenotype integrates with other aspects of stress signaling. Does this membrane association integrate with HSF1 signaling, and if so, does CHIP membrane association require HSF1?

4) Additional experiments to follow up on proteomic results are required. Are the membrane proteins identified as CHIP interactors ubiquitinated by this E3 ligase or does CHIP influence their activity through other mechanisms. There is no need to follow-up on all of the 'hits' but some follow-up on at least one hit seems appropriate.

5) All of the membrane-association images presented show few cells and are of rather poor resolution. These need to be improved. To what membranes does CHIP relocalize to? Are these Golgi membranes as the proteomics data would suggest? Double-label imaging experiments should be done to address this point.

6) As follow-up to comment 5, the authors should present statistics that indicate how many cells were imaged and how many showed the phenotypes they claim as 'representative'. This is the standard way to describe data of this sort.

7) From this reviewer's perspective, a better case can be made that it is PI4P that is the key sensing ligand for CHIP relocalization to membranes. The crude fat-blot experiments of Figure 2 and Figure 3 at least do not make a compelling case that PA is a regulatory lipid as excess chaperone does not strongly affect the binding. Yet the follow-up experiments focus on PLD. Perhaps PA strongly potentiates CHIP-binding to PI4P – that is, low PA enhances CHIP-binding affinity for PI4P. That possibility is consistent with all of the data and it can be tested in an appropriately-designed liposome flotation assay. Also, does knockdown/inhibition of π 4-OH kinase prohibit CHIP relocalization to intracellular membranes?

8) As follow-up to comment 7, PLD1 and PLD2 single and double knockout mice are viable and, curiously, these seem better protected against prion disease. This is not the expected outcome if one embraces the conclusion of this study. While this reviewer is not asking the authors to solve everything, the point is that maybe PA itself is a secondary factor with PI4P being the main player. If the data still point to PA, then the authors should at least comment on the mouse data in the context of their findings.

9) What happens to Golgi fragmentation is the CHIP-less MEFs are reconstituted with the m1 and m2 deletion forms of CHIP?

---

## [Author Response]

Summary:The authors propose the intriguing hypothesis that the Hsp70 associated E3 enzyme CHIP associates with specific membrane lipids to act as a "proteostasis sensor". The data supporting this role consist of intracellular localization of CHIP-EGFP under varying conditions, cell fractionation experiments, filter binding assays of CHIP to specific lipids, liposome binding experiments, and CHIP pull-down experiments coupled with MS. In general, the data are convincing that Hsp70 unassociated CHIP can associate with lipids and this association is related to membrane fragmentation caused by stress. The reviewers are in agreement that the study is interesting, breaks new ground, and appropriate for publication in eLife.

We would like to thank the Editor and the reviewers for appreciating the novelty and significance of our study. We hope that it will heighten the interest of cell biologists and biochemists for the relationship between cellular protein quality control, stress and architecture of mammalian cells.

However, the reviewers also agree that substantial gaps remain that can be ameliorated with some additional experimentation. Of particular concern is the functional relationship between CHIP relocalization and Golgi fragmentation.

We fully agree that revealing the mechanism of Golgi fragmentation during stress is a very interesting question with fundamental cell biological (e.g., additional functions of the Golgi) and pathophysiological (e.g., the role of the Golgi during neurodegeneration) implications. Excitingly, meantime we have acquired additional experimental evidence supporting the key finding of the study, namely, the functional relationship between chaperone availability and CHIP-induced reorganization of subcellular structure (Figure 5—figure supplement 1). Furthermore, prompted by the comments of the Editor and the reviewers, we have collected and are now including the following new data:

– Subcellular localization of CHIP during arsenite-induced stress (Figure 1—figure supplement 1).

– Binding of CHIP to liposomes containing 5% PI4P (Figure 2—figure supplement 1).

– Activation of HSF1 in the presence and absence of CHIP (Figure 5—figure supplement 1).

– Association of CHIP with cellular membranes when generation of PI4P is inhibited by wortmannin (Figure 5—figure supplement 1) or PI4KIIIb inhibitor (Figure 5—figure supplement Ff).

The reviewers also suggest another round of text editing (perhaps with assistance of colleague not directly involved in the work) to improve the readability and clarity of the manuscript.

An English native-speaker biochemist who was not involved in the study reformulated several passages in the text to improve the readability and clarity of the manuscript as suggested.

Essential revisions:1) It would be useful to quantify plasma membrane association for images other than the heat shock. For example, quantification of plasma membrane association of CHIP for images as shown in Figure 1 and Figure 1—figure supplement 1. Considering the subtlety of some of this phenotype (only ~20% of cells for the quantified heat shock), this is important to demonstrate the indicated effects.

The respective quantification panels are now included.

2) It would be interesting to see how universal this response is to other types of cytosolic stress. For example, As(III) induces protein misfolding in the cytosol and activation of the HSR. It would be interesting to see if As(III) induces a similar response.

The arsenite did not induce CHIP relocalization (Figure 1—figure supplement 1). Because arsenite causes protein damage via thiols, the oxidative damage protection might be the first line of cellular defense in this case. We are very thankful for this suggestion and included the result in the revised version because it implies specificity and argues against a trivial, i.e., general toxicity-driven, nature of CHIP binding to membranes.

3) It's unclear how the CHIP membrane association phenotype integrates with other aspects of stress signaling. Does this membrane association integrate with HSF1 signaling, and if so, does CHIP membrane association require HSF1?

Indeed, CHIP has been described as a component of the HSF1 activation machinery by Dai et al., (2003). Intuitively, the membrane relocalization (of a fraction) of CHIP during acute stress does not fit with the opposite direction of the HSF1 movement into nucleus. We analyzed the heat-induced HSF1 translocation into nuclei in wild-type and CHIP knock-out fibroblast and observed no differences in the time scale of the CHIP relocalization to membranes (Figure 5—figure supplement 1). It will be interesting to uncover in the future how distinct pools of cellular CHIP commit to different cellular compartments during stress response.

4) Additional experiments to follow up on proteomic results are required. Are the membrane proteins identified as CHIP interactors ubiquitinated by this E3 ligase or does CHIP influence their activity through other mechanisms. There is no need to follow-up on all of the 'hits' but some follow-up on at least one hit seems appropriate.

Ubiquitylation of new substrates on organelles would be the straight-forward explanation how chaperone-free CHIP can contribute to the reorganization of the cellular architecture during acute chaperone deficiency. Importantly, our analyses of the CHIP ligase activity *in vitro* showed that it remains preserved on liposomes.

In our study we focused on the initial (trigger) phase of CHIP mobilization to cellular membranes. As mentioned above, our proteomics results suggests the capacity of CHIP to interact with different organelles. Besides the Golgi apparatus, we have not investigated changes of other organelles yet. We believe that the mechanistic analysis of the effector phase (i.e., the cell biological consequences of the degradation of individual membrane-associated proteins) should go in parallel with a systematic assessment of subcellular structures. We kindly ask you to agree that this kind of in-depth analysis goes beyond the scope of the present focus.

5) All of the membrane-association images presented show few cells and are of rather poor resolution. These need to be improved.

Following the suggestion of the reviewers, we exchanged the problematic images of Figure 1—figure supplement 1. Arrow indications are not needed there anymore.

CHIP associates with membranes in a highly dynamic way. For example, the association is lost upon fixation of cells with a paraformaldehyde solution, a standard procedure before imaging. For this reason, live cell imaging was performed throughout. In doing so, we used low laser power to avoid phototoxicity and possible disruption of CHIP-membrane interaction. Secondly, we used the rather low 512x512 scanning format to increase the speed of image acquisition. Since only a fraction of cells reacted, we needed to acquire as many as possible cells to come up with statistically significant numbers. Cytosol vs. membrane localization was clearly distinguishable, so we applied the above measures as an acceptable trade-off.

To what membranes does CHIP relocalize to? Are these Golgi membranes as the proteomics data would suggest? Double-label imaging experiments should be done to address this point.

This is a difficult question to answer with a direct experimental evidence because of the highly dynamic nature of the interaction between CHIP and membranes as discussed above. The pull-downs of chaperone-free CHIP indicated an involvement of many different organelles (Figure 5). Whether these interactions are primary or rather related to the interaction with the Golgi apparatus has to be clarified with kinetic approaches using techniques capturing transient associations *in vivo* as, for example, mass spectrometry upon APEX (proximity labeling with ascorbate peroxidase).

6) As follow-up to comment 5, the authors should present statistics that indicate how many cells were imaged and how many showed the phenotypes they claim as 'representative'. This is the standard way to describe data of this sort.

We apologize for having scattered this information throughout the Materials and methods section in the previous version of the manuscript. We now listed all the numbers in one place at the end of the subsection Fluorescence microscopy.

7) From this reviewer's perspective, a better case can be made that it is PI4P that is the key sensing ligand for CHIP relocalization to membranes. The crude fat-blot experiments of Figure 2 and Figure 3 at least do not make a compelling case that PA is a regulatory lipid as excess chaperone does not strongly affect the binding. Yet the follow-up experiments focus on PLD. Perhaps PA strongly potentiates CHIP-binding to PI4P – that is, low PA enhances CHIP-binding affinity for PI4P. That possibility is consistent with all of the data and it can be tested in an appropriately-designed liposome flotation assay.

We would like to thank the reviewer for his/her interpretation of the blocking experiment in Figure 2 and Figure 3 and for suggesting the possibility of synergistic effects between PA and PI4P. PI4P is by far the most abundant phosphatidylinositol monophosphate species *in vivo*. We fully agree that it is the level of PI4P in membranes what could critically effect the final localization pattern of chaperone-free CHIP. We increased the PI4P concentration in liposomes from 1% to 5% and show now that PI4P can strongly enhance the association of CHIP with membrane (Figure 2—figure supplement 1).

We were careful not to claim the relative importance of PA over PI4P in our manuscript. We inhibited PA-generating PLDs primarily as a means to release the membrane-bound CHIP into the cytosol for further analyses. At the same time, the fact that the manipulation of either PA or PI4P (next paragraph) levels undocks CHIP suggests that both are important for CHIP localization *in vivo*.

Also, does knockdown/inhibition of π 4-OH kinase prohibit CHIP relocalization to intracellular membranes?

Following the suggestion of the reviewer, we assessed the relocalization of CHIP upon inhibition of the phosphoinositol-4 kinase with wortmannin (Figure 5—figure supplement 1) and PI4KIIIb-IN-10 (Figure 5—figure supplement 1). Wortmannin is a specific inhibitor of PI3K, however, it inhibits type III PI4K if used at higher concentrations. Both inhibitors reduced association of chaperone-free CHIP with membranes.

8) As follow-up to comment 7, PLD1 and PLD2 single and double knockout mice are viable and, curiously, these seem better protected against prion disease. This is not the expected outcome if one embraces the conclusion of this study. While this reviewer is not asking the authors to solve everything, the point is that maybe PA itself is a secondary factor with PI4P being the main player. If the data still point to PA, then the authors should at least comment on the mouse data in the context of their findings.

The pathophysiology of neurodegenerative diseases is complex and cellular stress response is only one of its components. For example, amyloid β-peptide can activate PLD2 and the lack of PLD2 reduces neuropathology in murine Alzheimer’s models. How cellular reorganization triggered by CHIP contributes to the amyloid toxicity in particular diseases remains to be determined. However, we fully agree with the reviewer regarding the possible importance of PI4P and have included a respective statement in the revised version of the manuscript in the Discussion section.

9) What happens to Golgi fragmentation is the CHIP-less MEFs are reconstituted with the m1 and m2 deletion forms of CHIP?

We focused on the m2 mutant, because the m1 mutant behaves as wild-type CHIP. The steady-state level of transfected CHIP-m2 mutant drops to ca. 20-30% of the wild-type protein level. Thus, the comparison of the effects on the Golgi fragmentation is not straightforward. One would need to transfect different amounts of the proteins risking additional effects. When transfected at equal amount as the wild-type CHIP, m2 mutant did not induce fragmentation of the Golgi apparatus (fraction of control 1.2 ± 0.1, N=3).